# Sero-prevalence of SARS-CoV-2 antibodies in Ethiopia: Results of the National Population Based Survey, 2021

**Geremew Tasew**[1], **Saro Abdella**[1], **Birra Bejiga**[1]*, **Jemal Ayalew**[2], **Masresha Tessema**[1], **Feyiso Bati**[1], **Abraham Ali**[1‡], **Legesse Negash**[1], **Enyew Birru**[1], **Getachew Tollera**[1‡], **Mohammad Ahmed**[1‡], **Adamu Tayachew**[1‡], **Dereje Nigussie**[1], **Laura Binkley**[3‡], **Joan-Miqel Balada-Llasat**[4‡], **Shu-Ha Wang**[4‡], **Leuel Lisanwork**[3], **Zelalem H. Mekuria**[3‡], **Hiwot Moges**[5‡], **Bernard Barekye**[5‡], **Marguerite Massinga Loembe**[5‡], **Mohamed Abdul Aziz**[5‡], **Eshetu Ejeta**[6‡], **Faiqa Kassim**[7‡], **Wondwossen Gebreyes**[4‡], **Abebaw Gebeyehu**[8‡], **Lia Tadsse**[8‡], **Dereje Duguma**[8‡], **Getnet Yimer**[3‡], **Desmond E. Williams**[9]

**1** Ethiopian Public Health Institute, Addis Ababa, Ethiopia, **2** Department of Statistics, Wollo University, Dessie, Ethiopia, **3** The Ohio State University Global One Health initiative, Addis Ababa, Ethiopia, **4** The Ohio State University Global One Health Initiative, Columbus, Ohio, United States of America, **5** Africa Centers for Disease Control and Prevention (ACDC), Addis Ababa, Ethiopia, **6** Ambo University, Ambo, Ethiopia, **7** World Health Organization (WHO), Addis Ababa, Ethiopia, **8** Federal Ministry of Health, Ethiopia, **9** U.S. Centers for Diseases Control and Prevention (CDC), Addis Ababa, Ethiopia

☯ These authors contributed equally to this work.
‡ This author contributed less but substantially.
* birr4allephi@gmail.com

## Abstract

### Background

SARS-CoV-2 pandemic has caused a continuing health crisis affecting the public health system globally. Population-based serological surveys are a highly valuable and recommended method to measure population exposure and spread of pandemic, given the existence of asymptomatic cases and little access to diagnostic testing. This national population-based study aims to estimate the seroprevalence of SARS-CoV-2 infection in all parts of Ethiopia and determine potential risk factors and burden of infection.

### Methods

A nationwide seroprevalence survey was done among 12,756 households (HHs) across the country using three-stage stratified sampling technique from April 15, 2021 to May 16, 2021 among population of Ethiopia above 15 years of age. One member of each of the selected HHs, who fulfilled the eligibility criteria, was randomly selected. We captured data using interviews and finger prick blood samples to test for anti-SARS-CoV-2 antibodies using high specificity rapid diagnostic tests (RDTs). A questionnaire was used to capture all necessary data on demographics, social exposure, and history of vaccination for SARS-CoV-2, symptoms compatible with SARS-CoV-2, and any known medical conditions. The data were collected using an open data kits (ODK) software and imported

**Data availability statement:** The data is available from the Ethiopian Public Health Institute data management center at https://ndmc.ephi.gov.et. Following a request sent to Birra Bejiga the EPHI Data Manager, at the e-mail address (birr4allephi@gmail.com) or at Fax: 011 275 8634. P.O. Box: 1242 or 5654 of Ethiopian Public Health Institute, Addis Ababa, Ethiopia. Other authors of this manuscript do not have any special privilege in accessing the data.

**Funding:** The author(s) received no specific funding for this work.

**Competing interests:** The authors have declared that no competing interests exist.

into STATA version 17 for analysis. Descriptive statistics (frequencies and proportions) were used to summarize data on the study variables. Forest plots and maps were used to visualize the seroprevalence of SARS-CoV-2 across various individual and environmental factors. The study sample was weighted, and the survey set command in Stata (svy) was used in the analyses to account for the survey design. Adjusted Odd ratio (AOR) was used to determine higher risk factors of having been infected at least once, 95% confidence interval to assess precision of the estimates, and a $P$ value $\leq$ 0.05 to determine statistically significant.

## Result and discussion

This study indicated the overall national prevalence of seropositivity was 9.3% that suggests nearly one in ten individuals in Ethiopia was exposed to SARS-CoV-2 infection by May 2021. All regional states in the country are affected with SARS-CoV-2 infection although infection was more common in densely populated regions. Seroprevalence was significantly higher among, individual, aged 35–44, 55–64 and 65 and over years had more odds of being infected by SARS-CoV-2 compared with those aged 15–24 years. The seroprevalence is also high among professional/technical occupations, and among those having at least one comorbidity. The participants who had seven and more members had higher odds of infection compared with those who had two or less members. The odds of infection among respondents, who reported having ever tested for COVID-19 and being sick since March 2020, were higher compared with their counterparts. Among the environmental factors, the odds of SARS-CoV-2 infection in urban residents were higher than in the rural setting. In relation to geographic administration boundaries, participants from Harari Region, Addis Ababa, and Benishangul Gumuz had higher odds of infection compared to those from Afar Regions respective.

## Conclusion and recommendations

This study reveals the overall seroprevalence of SARS CoV-2 antibodies in Ethiopia was 10.0% as of May 2021. The seroprevalence of IgG antibodies against COVID-19 is higher than that of IgM antibodies, indicating a past infection. SARS-CoV-2 antibody seroprevalence was varied by regional state, sex, residence area, age, and occupational status. It also suggests that the majority of Ethiopia's have inadequate knowledge of understanding about SARS-CoV-2 antibodies, we recommend strengthening public health and social measures to mitigate the spread of COVID-19 diseases, including increased vaccination coverage and testing capability. All responsible authorities and stakeholders working locally, nationally, and globally need to support strengthening health systems and be prepared to combat morbidity and mortality and to encourage ongoing vaccination efforts. Periodic seroprevalence surveys will aid in monitoring the status and progress of the COVID-19 pandemic.

## Introduction

A novel coronavirus, severe acute respiratory syndrome coronavirus 2 (SARS-CoV-2), was identified in December 2019 in China [1], and caused a pandemic of respiratory illness known as coronavirus disease 2019 (COVID-19) [2]. It infects people of all ages and sexes, and

individuals in older age groups and those with underlying medical conditions are at a higher risk of getting severe COVID-19 disease [3,4].

Most countries to date have focused on testing individuals with symptoms or in high-risk groups with RT-PCR testing while several studies have reported that a substantial fraction of infected individuals develop mild symptoms or even remain asymptomatic while, the number of SARS-CoV-2 infections is an underestimation of the true number of infections [4–7]. In addition, even for symptomatic cases, access to testing, diagnosis and reporting could be limited, Although the algorithm used for RT-PCR testing can help in monitoring and preventing the spread of the virus, there are several limitations when used alone [8–10]. First, mild or asymptomatic cases are unscreened and second, in most cases the PCR testing only detects positives if the patient currently has the virus in the body, in a window of about 10 days from infection [11]. After 10 days of symptom onset, the PCR sensitivity of the test is estimated at only 67% and decreases further over time [11]. This testing therefore cannot detect individuals that have previously been infected and fought off the disease [11]. The number of people that showed no or mild symptoms is unknown and could be 50 to 100 times higher, or more, than the reported number of cases [8,12].

Due to these limitations and the need to estimate the proportion of the population infected, an important measure for public health decision making regarding population-level infection risk, several countries have begun testing a wider sample of people using serological tests [13–15]. Serological tests, which detect antibodies produced by specialized white blood cells (B cells), help to estimate the true extent of the burden of COVID-19 infections in the country and assist efforts to curb the pandemic [16–18].

Serological surveys in a population are a valuable tool to conduct comprehensive assessments of exposure and spread of the pandemic, given the existence of asymptomatic cases and little access to diagnostic tests [4]. An antibody response occurs in symptomatic or asymptomatic SARS-CoV-2 infected individuals. The corresponding serological markers may persist for weeks to months; the neutralizing activity of the antibodies may also increase over time while the sensitivity of PCR testing decreases over time [6,19].

Most African countries including Ethiopia have been relatively spared from the devastating effect of the pandemic to date [20]. This is likely due to multiple contributing factors, one of which may be a large number of untested/unreported cases [20,21]. Other potential reasons are the relatively young median age of the population who are less prone to develop severe forms of the disease and stay undetected at home, lower rates of transmission owing to the effectiveness of early confinement measures such as border closures, high levels of nonspecific immunity and other potential factors [20,21]. Apart from this, considerable uncertainty regarding the evolution of the disease, limited testing capacity, inefficient surveillance and contact tracing, and inability to detect the circulation of variants of concern could be the other contributing factors to underestimating the burden of the pandemic in the country [22,23]. For instance, in Ethiopia a population of more than 115 million, only 52,131 positive cases were reported as of August 31, 2020 out of 910,293 people tested for SARS-CoV-2[12]. This shows a relatively low case rate compared to other countries [12].

In this context, SARS-CoV-2 seroprevalence surveys accompanied by epidemiological data on symptoms, risk of exposure and demographics could inform estimates of the proportion of the population that has already developed antibodies against SARS-CoV-2, vaccination campaigns, model future scenarios of spread, and provide the necessary information to policy makers looking to balance necessary social interaction and movement of individuals with the containment of COVID-19.

The aim of this population-based seroprevalence study is to estimate the proportion of the population that has antibodies against SARS-CoV-2, assess the geographic distribution of

SARS-CoV-2 antibodies in Ethiopia, and determine the associated socio-demographic factors associated with seropositivity in Ethiopia. This will allow policy makers and stakeholders to understand the disease burden, monitor disease trends, and strategize for future needs. In addition, this survey could help to identify the most vulnerable populations and regions to target for risk mitigation measures such as vaccination, to curb viral transmission, reducing critical illness and death.

## Methods and materials

### Study area and design

We conducted the national SARS-CoV-2serosurvey from April 15, 2021 to May 16, 2021. This serosurvey was designed to cover all regions of Ethiopia, and both rural and urban areas. At the time, the country is divided into ten geographical regions and two city administrations. Each region is sub-divided into zones and regional towns, each zone sub-divided into woredas (districts) and towns, each woreda divided into woreda towns and kebeles (the lowest administrative unit), and all towns at all levels into kebeles. A three-stage cluster, population-based, cross sectional study design was used to estimate the proportion of SARS-CoV-2 infection in the selected census enumeration areas (EAs). The source population was the entire population of Ethiopia above 15 years of age; the study population was all population residing in the selected EAs during the study period. Individuals whose age was less than or equal to 15 years, those who were critically ill, and those who were not willing to consent were excluded.

**Sample size and sampling procedure.** A three-stage stratified cluster random sampling was used to select representative sample of population. The sampling frame of this study includes all census enumeration areas (EAs) created for the 2019 Ethiopia Population and Housing Census (EPHC) by the Central Statistical Agency (CSA). An EA is a geographic area covering an average of 131 households (HHs). The sampling frame contains information about EA location, type of residence (urban or rural), and estimated number of residential HHs. Each region was stratified into urban and rural areas. Samples of EAs were selected independently in each stratum

In the first stage, EAs were used as the primary sampling unit (PSU) and 462 EAs (208 EAs in urban areas and 254 in rural areas) were selected proportionally to the EA size, where size is the number of households in each EA. A household (HH) listing operation was carried out in all the selected EAs, and the resulting lists of HHs served as sampling frame for the selection of HHs in the second stage.

In the second stage of selection, a fixed number of 30 HHs per selected EA were randomly selected with an equal probability from the newly prepared HH listing. The lists will consist of enumeration areas prepared for all regions. From the lists prepared, a predetermined number of enumeration areas, which contain, 509 EAs were selected.

In third stage, one member of the HH with eligibility criteria was randomly selected from each selected HHs. Sample size calculation was done using design factor approach with assumption of expected prevalence of 7% from the program data; the design factor for different stage sampling scheme (deft = 1.5), the margin of error ($\delta$ = 0.048) and non-response rate (1-r) which yields (r = 0.1) by taking into account an 10% non-response rate in the study area which provided a 10% non-response rate contingency. The final minimum sample size for this study was 13,286.

Hence, the national required minimum sample size is computed using the following sampling formula.

$$n = \frac{deft^2 \left( \frac{1}{p} - 1 \right)}{(1-r)\delta^2}$$

Where;

- *Deft* is the design (effect) factor for different stage sampling scheme, 1.5 has been used.

- P is the expected value of prevalence. The current national prevalence from the program data is7%.

- δ is the relative precision/margin of error and value of 4.8% was used.

- r is the non-response rate which is assumed to be 10%in the study area. This is the amount of drop out and its complement (1-r). Therefore, 1-r is response rate among the study participants and used to adjust for dropouts.

Based on the given formula and the aforementioned inputs, the required sample size used for this assessment was 13,286households.

## Data collection and participant consent

A total of 154 field enumerators, 41 field supervisors, 24 regional coordinators and 4 central coordinators were involved in the survey activity. Guidelines were prepared for selection of sample HHs and data collection using an interviewer-administered questionnaire.

Data collectors and field supervisors were health professionals in public health and Laboratory trained and evaluated on ethical considerations, data collection methods, and protocols for performing the RDT for COVID-19 before they deployed to undertake their field activities. RDT testers were trained on how to use the test kit and control lines. All invalid results were repeated. Quality control (QC) panels consisting of a positive and negative control specimen were done in parallel with the testing procedure to ensure test kits were performing correctly. The field-level sample selection activities were quality-controlled and checked by survey management staff (survey coordinators, and/or trained supervisors) during the data collection process. Professional laboratory technologists were in charge of the collecting of blood samples, the assay process, the interpretation of the test results, and waste disposal management.

Every HH was provided with information about the nature of the study and all participants gave their written informed consent before participating. For individuals younger than 18 years, a parent or legal representative provided informed consent while the child also provided an assent. Participants were then tested with a RDT and asked to respond to a short questionnaire to capture data on demographics, social exposure history, symptoms compatible with COVID-19, any known medical conditions, previous diagnosis of COVID-19, and previous vaccination against SARS-CoV-2. Answers and test results were collected on a tablet using ODK (Open Data Kit) [24]. Institutional Review Board (IRB) of Ethiopian Public Health Institute (EPHI) ethically approved the study.

## Serology testing

Anti-SARS-CoV-2 antibodies using high specificity rapid diagnostic tests were used because it could be performed in the field and reduced the complexity and cost of the survey since it eliminated the need for specimen transport and provided quick results. The CE approved Canea COVID-19 IgM/IgG rapid test (Core Technology Co., Ltd, Beijing, China) for detection of IgM (Immunoglobulin M) and IgG (Immunoglobulin G) to SARS-CoV-2 infection because was used. Ethiopian Food, Medicine, and Health Care Administration and Authority (FMHACA) had approved this kit for use in Ethiopia. Positive samples were sent to the Ethiopian Public Health Institute (EPHI) and confirmed by using a polymerase chain reaction (PCR). The test requires finger prick capillary blood samples, which are easily collected in

rural sites far from sample storage and processing facilities. The test manufacturer reports a sensitivity of 97.6% (94% for IgM only and 94% for IgG only) and specificity of 100% was used for this study. The RDT is a lateral flow immunoassay that qualitatively detects and differentiates IgM and IgG antibodies against SARS-CoV-2 in whole blood specimens, results are produced and read in 10–15 minutes according to the manufacturer.

## Statistical analysis

Data were collected electronically via ODK and exported to STATA version 17 for analysis(15).

Frequencies, percentages, and the data were summarized and presented in tables using 95% confidence intervals (CIs). Forest plots and maps were used to show Covid-19 seroprevalence in relation to individual and environmental factors. The study sample was weighted, and the survey set command in Stata (svy) was used in the analyses to account for the survey design.

To account for the survey design effect in identifying associated factors, including individual and environmental-level variables, we employed a two-level multilevel mixed-effects logistic regression model. All the independent variables categorized as individual-level variables were considered as level-1 variables and the primary sampling unit (PSU) (Enumeration Area (EAs)) as level-2 variables. We used intra-class correlation (ICC) to quantify the effect of the level-2 variables (EAs) in the multilevel regression model, and the between-cluster variation accounted for the proportion of total variation in the response variable. It was followed with a bivariate analysis using Pearson's chi-square (X2) to evaluate the relationships between seroprevalence and each of the study's explanatory factors. Statistically significant variables (with p-values < 0.05) in the bivariate analysis were included in the multilevel logistic regression model. Adjusted odds ratios (AORs) with 95% confidence intervals (CIs) were used to present the results of the multilevel regression.

In addition, we fitted a series of models and selected the best model for the main analysis. The first model (Model 0) was the intercept-only model with no explanatory variable. The second model (Model 1) included the individual variable and Model 2 included only environmental variables. The final model (Model 3) was the full model that included both the individual-level and environmental-level variables. Stata command "melogit" was used in fitting these four models (Models 0–3). Model comparison was done using the Akaike's Information Criterion (AIC). All results were presented as adjusted odds ratios (AOR) with 95% CI; those AOR did not include one declared as significant variables.

## Ethical consideration

Ethical approval for the study protocol was obtained from the Scientific and Ethical Research Office (SERO) of the Ethiopian Public Health Institute (EPHI). Potential participants were told about the study purpose and procedures, potential risks, and protections using the local language. Written informed consent was obtained from each survey participant for the interview and blood sample collection, and for testing.

## Results

A total of 12,756 HHs, (96%) of the sample size of 13,286 across Ethiopia participated in the seroprevalence survey conducted from April 15 to May 16, 2021. The test results were invalid for 10 participants and these individuals were excluded from further analyses. A total of 4,996 men and 7,750 females were included in the analyses. The mean age of the sample population was 34.8 years (Table 1).Four hundred and seventy-nine individuals (2.6%) reported receiving at least one dose of a SARS-CoV-2 vaccine before testing. Vaccinated individuals were

**Table 1. Demographic and general characteristics and seropositivity in Ethiopia, May2021.**

| | Total Population | Pop. Distribution | Population (unvaccinated) | Seroprevalence (%) | 95% CI |
|---|---|---|---|---|---|
| **Antibody Positivity** | | | | | |
| Negative | 17479047 | 89.8 | | | |
| Positive | 1977313 | 10.2 | | | |
| **Received COVID-19 Vaccine?** | | | | | |
| Yes | 504555 | 2.6 | 208961 | 41.4 | 33.6–49.7 |
| No | 18961357 | 97.4 | 1768352 | 9.3 | 8.2–10.6 |
| **Sex** | | | | | |
| Male | 8291642 | 43.8 | 775306 | 9.4 | 7.9–11.1 |
| Female | 10660163 | 56.2 | 993046 | 9.3 | 8.1–10.7 |
| **Age**** | | | | | |
| 15–24 | 3966341 | 20.9 | 337298 | 8.5 | 7–10.3 |
| 25–34 | 6493313 | 34.3 | 540684 | 8.3 | 7–9.9 |
| 35–44 | 4080387 | 21.5 | 404559 | 9.9 | 8.3–11.9 |
| 45–54 | 2070124 | 10.9 | 219082 | 10.6 | 8–13.9 |
| 55–64 | 1330351 | 7 | 169231 | 12.7 | 9.1–17.6 |
| 65 + | 1011289 | 5.3 | 97498 | 9.6 | 6.9–13.3 |
| **Urban/Rural**** | | | | | |
| Urban | 4810084 | 25.4 | 623358 | 13 | 11.5–14.6 |
| Rural | 14141721 | 74.6 | 1144994 | 8.1 | 6.8–9.7 |
| **Region**** | | | | | |
| Tigray | 123300 | 0.7 | 13160 | 10.7 | 6.5–17.1 |
| Afar | 432650 | 2.3 | 21052 | 4.9 | 3.1–7.7 |
| Amhara | 4799096 | 25.3 | 430593 | 9 | 6.7–11.8 |
| Oromiya | 7313338 | 38.6 | 591057 | 8.1 | 6.3–10.2 |
| Somalia | 1010985 | 5.3 | 67097 | 6.6 | 4.2–10.4 |
| Benishangul Gumuz | 225052 | 1.2 | 25196 | 11.2 | 8.2–15.1 |
| SNNP | 3035900 | 16 | 330865 | 10.9 | 8–14.6 |
| Sidama | 841542 | 4.4 | 81063 | 9.6 | 7.3–12.7 |
| Gambella | 96736 | 0.5 | 5760 | 6 | 4–8.8 |
| Harari | 60703 | 0.3 | 14772 | 24.3 | 18.7–31 |
| Diredawa | 110009 | 0.6 | 12737 | 11.6 | 8.8–15.1 |
| Addis Ababa | 902494 | 4.8 | 175000 | 19.4 | 15.8–23.6 |
| **Previous COVID-19 test**** | | | | | – |
| Yes | 129153 | 0.7 | 19601 | 15.2 | 9.6–23.2 |
| No** | 18822652 | 99.3 | 1748751 | 9.3 | 8.2–10.5 |
| **Occupation** | | | | | |
| Agriculture | 7913057 | 41.8 | 580907 | 7.3 | 5.9–9.1 |
| Housewife | 5990809 | 31.6 | 667376 | 11.1 | 9.4–13.1 |
| Healthcare worker | 90991 | 0.5 | 7350 | 8.1 | 3.7–16.7 |
| Professional/technical/ managerial | 427099 | 2.3 | 59901 | 14 | 9.9–19.5 |
| Public-facing sales and services | 935382 | 4.9 | 108734 | 11.6 | 9.2–14.6 |
| Skilled/Unskilled manual | 892487 | 4.7 | 85685 | 9.6 | 7.1–12.9 |
| Student (10 or 20) | 1591506 | 8.4 | 134490 | 8.5 | 6.3–11.2 |
| Unemployed | 687271 | 3.6 | 79369 | 11.5 | 8.9–14.8 |
| Other | 423203 | 2.2 | 44540 | 10.5 | 6.6–16.5 |
| **No of household residents** | | | | | |
| ≤ 2 | 5175998 | 27.3 | 456788 | 8.8 | 7.3–10.6 |

*(Continued)*

**Table 1.** (Continued)

| | Total Population | Pop. Distribution | Population (unvaccinated) | Seroprevalence (%) | 95% CI |
|---|---|---|---|---|---|
| 3–5 | 9333915 | 49.3 | 841866 | 9 | 7.7–10.6 |
| 7 + | 4441892 | 23.4 | 469698 | 10.6 | 8.7–12.8 |
| **Smoking** | | | | | |
| No | 18273672 | 96.4 | 1712604 | 9.4 | 8.2–10.6 |
| Yes | 678133 | 3.6 | 55748 | 8.2 | 4.8–13.7 |
| **Have you been sick?**\*\* | | | | | |
| Never | 16464836 | 86.9 | 1312264 | 8 | 6.9–9.2 |
| At least once | 2337125 | 12.3 | 426666 | 18.3 | 15.4–21.6 |
| At least twice | 149844 | 0.8 | 29422 | 19.6 | 10.3–34.1 |
| **Pre-existing medical condition**\*\* | | | | | – |
| Not have | 17260609 | 91.1 | 1556550 | 9 | 7.9–10.3 |
| At least one | 1593948 | 8.4 | 211091 | 13.2 | 10.9–16 |
| Don't know | 97248 | 0.5 | 711 | 0.7 | 0.1–9.1 |

\**Analyses restricted to the unvaccinated population.* \*\**p < 0.05* The national seroprevalence of IgG and IgM antibodies to SARS-CoV-2 in the unvaccinated population was 9.3% (95% CI: 8.2–10.6). In this population, the prevalence of IgM was 2.8%, IgG 4.7%, and IgG and IgM combined was 1.9% (Table 2). A study participant was considered antibody positive if they had any IgM, IgG, or IgG and IgM combined.

excluded from this study. Since this survey was primarily designed to ascertain the prevalence of SARS-CoV-2 antibodies due to prior SARS-CoV-2 infection in the community, we excluded these 479 individuals in subsequent analyses (Table 1).

The seroprevalence by demographic characteristics are presented in (Table 1). Seroprevalence was similar in both males, 9.4% (CI 7.9–11.1) and females, 9.3% (CI 8.1–10.7). The prevalence varied significantly by age category and was highest in those 55–64 years, 12.7 (CI 9.1–17.6) and lowest among participants 25–34 years, 8.3% (CI 7.0–9.9). Rural residents had a significantly lower seroprevalence 8.1% (CI 6.8–9.7) than urban residents 13.0% (CI 11.5–14.6) Table 1.

The seroprevalence also varied by region with the highest seroprevalence in Harari, 24.3% (CI 18.5–31.3) and Addis Ababa, 19.4% (CI 15.7–23.7) and the lowest in Afar, 4.9% (CI 3.0–7.8) and Gambella 6.0% (CI 3.9–8.9). The seroprevalence in each region is graphically represented in (Fig 1), maps of the seroprevalence in the woredas of Harari are shown in (Fig 2), and the sub-cities of Addis Ababa in (Fig 3).

The most common occupation among the survey participants was agricultural worker (41.8%) and housewife (31.6%) and only 0.5% were healthcare workers. Seroprevalence was significantly different by occupation (p < 0.05), with the highest among technical and managerial staff, 14.0% (CI 9.9–19.5) and lowest in agricultural workers, 7.3% (CI 5.9–9.1). Seroprevalence among healthcare workers was 8.1% (CI 3.7–16.7) Table 1.

As indicated in Table 1. Seroprevalence increased with the number of HH residents, but the difference was not significant. Seroprevalence was not significantly different between smokers,

**Table 2. Distribution of IgG, and IgM antibodies to SARS-CoV-2 virus in the unvaccinated population in Ethiopia, May 2021.**

| SARS-CoV-2Antibody test result | Frequency | Weighted Proportion (95%CI) |
|---|---|---|
| IgM (+) | 403 | 2.8(2.3, 3.5) |
| IgG (+) | 832 | 4.7(4.0, 5.4) |
| IgM (+) and IgG (+) | 364 | 1.9 (1.5, 2.4) |
| IgM (-) and IgG (-) | 11,147 | 90.7(89.4, 91.8) |

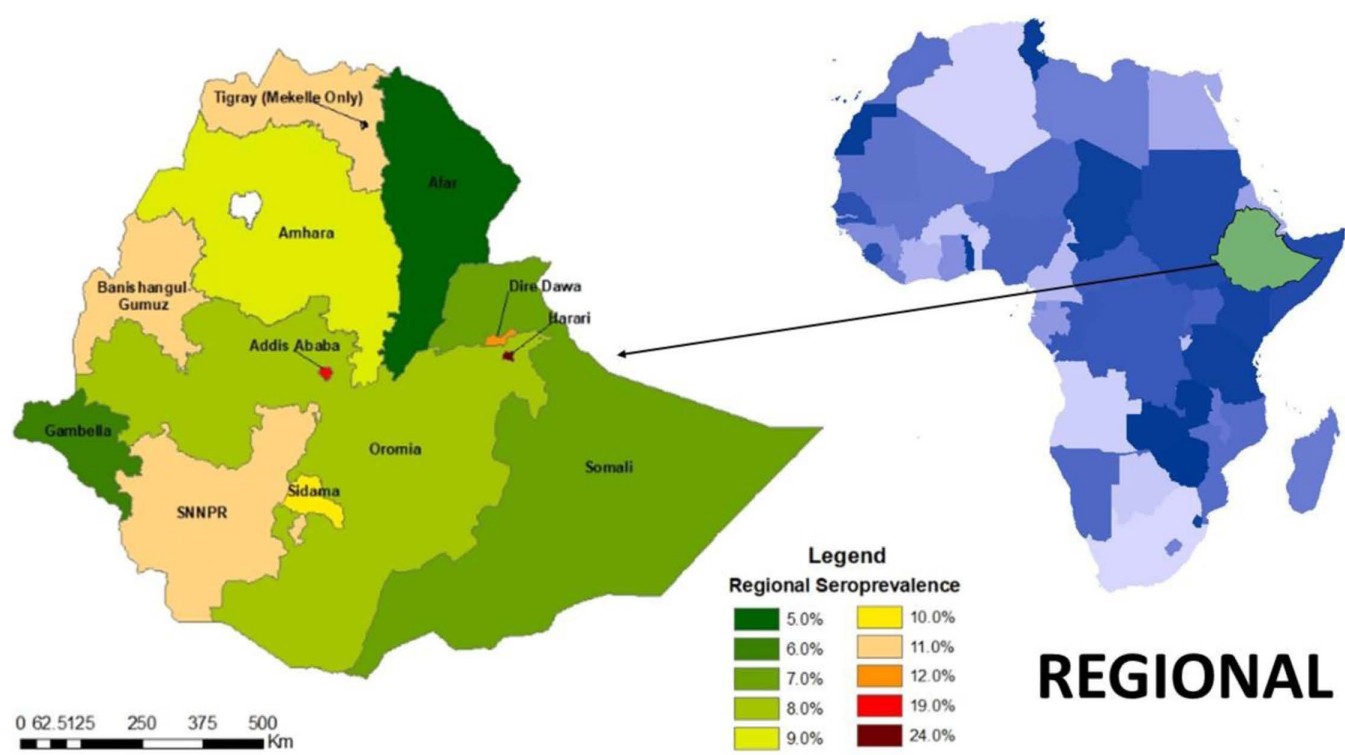

**Fig 1. Seroprevalence of SARS-CoV-2 Antibodies in Ethiopia by region, May 2021.**

8.2% (CI 4.8–13.7) and non-smokers, 9.4% (CI 8.2–10.6). Seroprevalence was significantly higher in those who reported being sick once or twice in the period over the last two months. Those who were sick once, 18.3% (CI 15.4–21.6) or twice, 19.6% (CI 10.3–34.1) during this period had a much higher prevalence than those who did not report being sick, 8.0% (CI 6.9–9.2) Table 1 Participants who had at least one pre-existing condition had a high seroprevalence 13.2% (CI 10.9–16.0) than those without pre-existing conditions 9.0% (CI 7.9–10.3). Study participants who reported a previous test for COVID-19 had a significantly higher seroprevalence, 15.2% (CI 8.5–21.8) than those who had not previously been tested, 9.3% (CI 8.1–10.5) Table 1. Additional adjustments for age and sex did not change the seroprevalence estimates or significance of these characteristics.

### Factors associated with SARS-CoV-2Seroprevalence results of a multilevel mixed effect logistic regression model

Results of the multilevel mixed-effect logistic regression model analyses are presented in (Table 3). In the multilevel analysis full model (model 3), which includes both individual and environmental variables, fitted the data better than the other candidate models, which had the lowest AIC, and was chosen as the final model for analysis for examining the associated factors for SARS-CoV-2seroprevalence. In the intercept-only model (Model 0), were substantial variations in the likelihood of the seroprevalence of SARS-CoV-2across the clustering of the PSUs (EAs) [$\sigma^2 = 1.00$, 95% CI 0.79–0.1.25]. The ICC varied between 17% and 23% for all models; the full model (Model 3) accounted for 17%, while the intercept-only model (Model

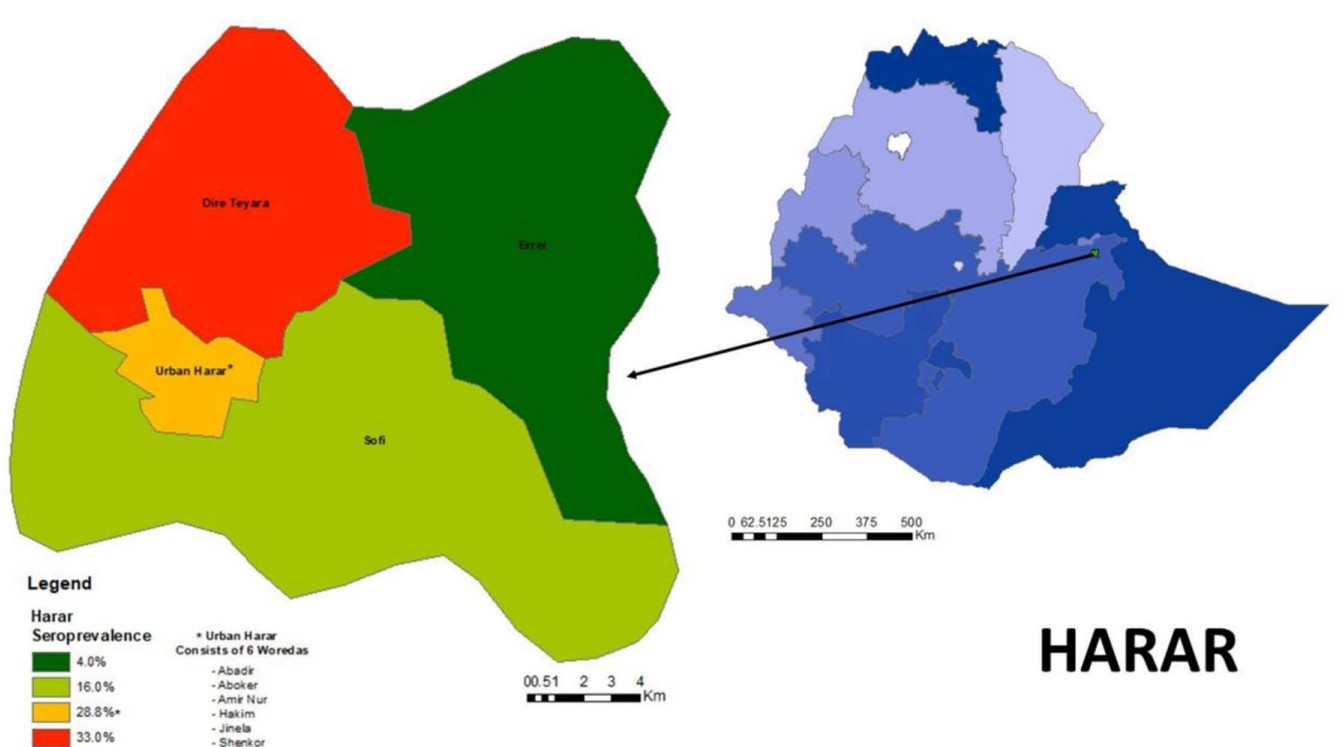

**Fig 2. Seroprevalence of SARS-CoV-2 Antibodies in Woredas in Harari region, Ethiopia, May 2021.**

0) accounted for 23%. As a result, the final model revealed the variation explained about 17% of the livelihood of SARS-CoV-2seroprevalence among the EAs in Ethiopia. As shown by the 95% CI (0.14, 0.21), the variation among EAs is statistically significant; thus, any analysis that does not account for this effect will result in a biased estimate (Table 3).

Based on the final model, age, number of household members, occupation, previous history of COVID-19 testing, and ever been sick since March 2020 were identified as individual significant associated factors, whereas region and current residence were identified as environmental factors (Table 3).

Study participants who were, 35–44 years old (AOR = 1.33; 95% CI = 1.1, 1.67), 45–54 years old (AOR = 1.62; 95% CI = 1.28, 2.05), 55–64 years old (AOR = 1.78; 95% CI = 1.35, 2.35), and 65 years and older (AOR = 1.64; 95% CI = 1.21, 2.24) had higher odds of having COVID-19 infection than those who were 15–24 years old.

The risks of infection were higher for study participants with seven or more household members than for those with two or fewer (AOR = 1.23; 95% = 1.02, 1.48). The odds of infection among respondents reported ever tested covid-19 and being sick since March 2020 were 1.59 and 2.38 times higher compared with their counterparts (AOR = 2.38; 95% = 2.03, 2.79, AOR = 1.59; 95% = 1.01, 2.52),respectively (Table 3).

When it came to environmental factors, participants who lived in urban areas had 1.5 times greater odds of contracting SARS-CoV-2infection than those who lived in rural areas (AOR = 1.5; 95% CI = 1.18, 1.9). Participants from the Harari region, Addis Ababa, and Benishangul Gumuz had 5.79, 2,91, and 2.74-times higher odds of infection compared to those from the Afar region, respectively, with respect to regional administrative limits (Table 3).

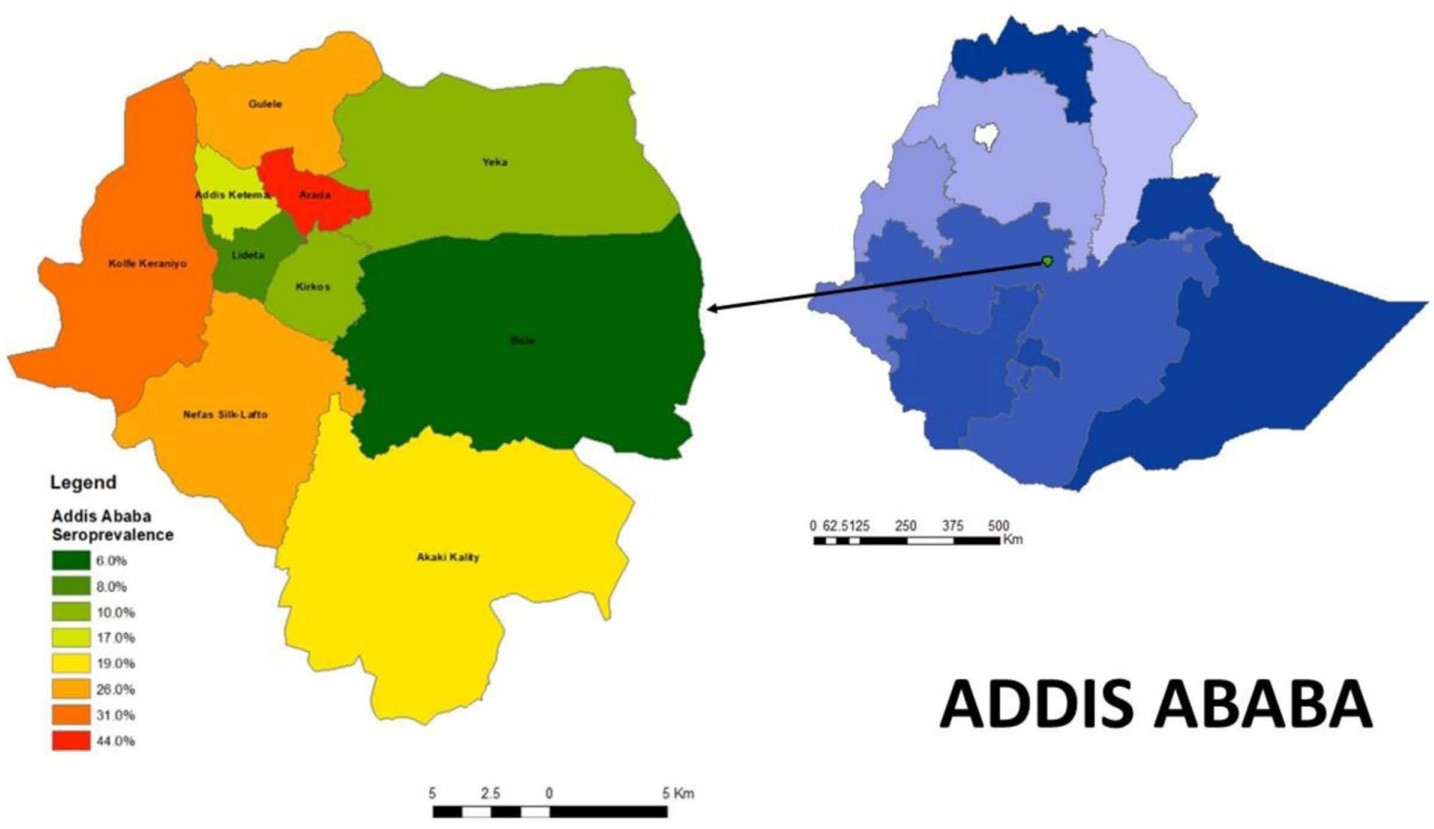

**Fig 3. Seroprevalence of SARS-CoV-2 Antibodies in Sub cities in Addis Ababa, Ethiopia, May 2021.**

## Discussion

SARS-CoV-2 seroprevalence is essential to assess the potential immunity in a population [25], to know the extent of the population exposure [26], and aid in the implementation of informed infection control and prevention strategies [27].

Hence, in this nationwide study, the seroprevalence of anti-SARS-CoV-2 antibodies in the unvaccinated Ethiopian population during the time period of April 15, 2021–May 16, 2021 was 9.3% (CI 8.2–10.6). The seroprevalence of anti-SARS-CoV-2 antibodies is higher than a study conducted in Addis Ababa Ethiopia, 3% (CI 6.0–8.6) in 2020 [28].

Moreover, the seroprevalence documented by this study is less than study done in Jordan, 35.2% (95% CI: 37.5–38.5%) [29], in Brazil, 11.9% (95% CI 10.7–13.1) [30], in Iran, 17.1% (CI 14.6–19.5) [31], in California, 11.5% (95% CI: 10.5–12.4) [32], in Zambia, 11% (CI 10–12) [33], in Egypt, 35% (CI 32–37) [34] and in South Sudan, 38% (CI 36–41) [35].

The possible reason for the observed variation in seroprevalence might be due to differences in socio-demographics, timing of the study, prevalence of SARS-CoV-2, compliance with and enforcement of non-pharmaceutical interventions against COVID-19, sample size, sampling technique, methods for serological testing, and sensitivity and specificity of serological tests.

Our study showed that there was no statistically significant difference in seroprevalence among male and female subjects. This finding is consistent with other population-based studies in Spain [4], Brazil [36], France [36], and India [37]. However, other studies showed higher seroprevalence among males than females [26,38,39].

**Table 3. Multilevel mixed effect logistic regression model findings for the predictors of SARS-CoV-2 in Ethiopia, 2021.**

| Variables | Model 0 | Model 1 | Model 2 | Model 3 |
|---|---|---|---|---|
| | AOR(95% CI) | AOR(95% CI) | AOR(95% CI) | AOR(95% CI) |
| *Individual level factors* | | | | |
| Age | | | | |
| 15–24 | | 1 | | 1 |
| 25–34 | | 1.05(0.87, 1.28) | | 1.04(0.86, 1.26) |
| 35–44 | | 1.39(1.13, 1.71) | | 1.36(1.1, 1.67) |
| 45–54 | | 1.69(1.33, 2.14) | | 1.62(1.28, 2.05) |
| 55–64 | | 1.84(1.40, 2.43) | | 1.78(1.35, 2.35) |
| 65 + | | 1.71(1.25, 2.32) | | 1.64(1.21, 2.24) |
| Occupation | | | | |
| Agriculture | | 1 | | 1 |
| Housewife | | 1.71(1.42, 2.06) | | 1.57(1.29, 1.91) |
| Healthcare/Professional/ technical/ managerial | | 1.57(1.15, 2.15) | | 1.33(0.97, 1.83) |
| Public-facing sales and services | | 1.51(1.16, 1.97) | | 1.35(1.02, 1.77) |
| Skilled/Unskilled manual | | 1.62(1.08, 2.43) | | 1.37(0.91, 2.06) |
| Student (10 or 20) | | 1.2(0.93, 1.55) | | 1.06(0.82, 1.38) |
| Unemployed | | 1.7(1.27, 2.28) | | 1.53(1.13, 2.06) |
| Other | | 1.28(0.85, 1.94) | | 1.17(0.77, 1.78) |
| No of household members | | | | |
| ≤ 2 | | 1 | | 1 |
| 3–6 | | 0.98(0.85, 1.14) | | 0.99(0.85, 1.14) |
| 7 + | | 1.14(0.95, 1.37) | | 1.23(1.02, 1.48) |
| Ever been sick since March 2020. | | | | |
| No | | 1 | | 1 |
| Yes | | 2.44(2.08, 2.86) | | 2.38(2.03, 2.79) |
| Ever tested for COVID-19 | | | | |
| No | | 1 | | 1 |
| Yes | | 1.63(1.03, 2.59) | | 1.59(1, 2.52) |
| *Environmental level factors* | | | | |
| Region | | | | |
| Afar | | | 1 | 11 |
| Tigray | | | 1.48(0.70, 3.11) | 1.73(0.82, 3.63) |
| Amhara | | | 1.81(1.07, 3.07) | 1.93(1.13, 3.27) |
| Oromiya | | | 1.71(1.01, 2.88) | 1.82(1.08, 3.08) |
| Somalia | | | 1.25(0.7, 2.21) | 1.11(0.63, 1.98) |
| Benishangul Gumuz | | | 2.43(1.37, 4.31) | 2.74(1.54, 4.88) |
| SNNP | | | 2.22(1.31, 3.75) | 2.28(1.34, 3.87) |
| Sidama | | | 2.00(1.14, 3.51) | 2.14(1.22, 3.76) |
| Gambella | | | 1.16(0.63, 2.12) | 1.23(0.67, 2.26) |
| Hareri | | | 5.57(3.16, 9.82) | 5.79(3.28, 10.21) |
| Diredawa | | | 2.08(1.17, 3.72) | 2.08(1.16, 3.72) |
| Addis Ababa | | | 3.31(1.94, 5.65) | 2.91(1.70, 4.97) |
| Residence | | | | |
| Rural | | | 1 | 1 |
| Urban | | | 1.6(1.27, 2) | 1.5(1.18, 1.9) |
| Constant | | 0.04(0.03, 0.06) | 0.04(0.02, 0.06) | 0.02(0.01, 0.03) |

*(Continued)*

**Table 3.** (Continued)

| Variables | Model 0 | Model 1 | Model 2 | Model 3 |
|---|---|---|---|---|
| | AOR(95% CI) | AOR(95% CI) | AOR(95% CI) | AOR(95% CI) |
| Random intercept model | | | | |
| PSU variance (95% CI) | 1.0(0.79, 1.25) | 0.92(0.73, 1.17) | 0.70(0.54, 0.90) | 0.69(0.53, 0.89) |
| ICC | 0.23(0.19, 0.28) | 0.22(0.18, 0.26) | 0.17(0.14, 0.21) | 0.17(0.14, 0.21) |
| Wald chi-square (p-value) | Ref | 386.72 (<0.001) | 100.72 (<0.001) | 312.43 (<0.001) |
| Model fitness | | | | |
| Log-likelihood | −4078.34 | −3964.13 | −4030.977 | −3922.54 |
| AIC | 8160.675 | 7964.262 | 8089.953 | 7905.074 |
| BIC | 8175.504 | 8097.726 | 8193.759 | 8127.515 |
| N | 12,267 | 12,267 | 12,267 | 12,267 |
| Number of groups | 436 | 436 | 436 | 436 |

Seroprevalence also varied markedly by region and type of residence, with the highest seroprevalence found in Harari regional state, Addis Ababa city administration, and in urban residents. Although the testing per capita by the surveillance system has been extremely low outside of Addis these estimates confirmed the patterns observed from the surveillance data, which recorded higher rates of infection in densely populated regions like Addis and Harari and much lower rates of transmission in sparsely populated regions such as Afar and Gambella.

According to studies done in Nigeria and India, seroprevalence was higher among those residing in urban areas than in rural areas [37,40]. The study finding is also in line with previous findings that seroprevalence was higher in more densely populated areas. The overall seroprevalence from New York State in the USA was 12.5%, but in New York City where population density is higher, the estimated seroprevalence was 22.7%; overall seroprevalence in Kenya of 5.6% was lower than in Mombasa, 8.0% and 7.3% in Nairobi [4,39,41].

Participants who had at least one pre-existing condition had a higher seroprevalence, 13.2% (95% CI 10.9–16.0) than those without pre-existing conditions, 9.0% (95% CI 7.9–10.3), p < 0.05. This finding is similar to studies reporting higher seroprevalence among those with predisposing factors [42].

Regarding occupation, the highest seroprevalence was among professional/technical/managerial staff, and the lowest in agricultural workers. In this general population study, we found that healthcare workers were not the most affected group, which is similar to previous general population study findings [28,34,40].

Moreover, in this study we found that the seroprevalence of anti-SARS-CoV-2 antibodies varied significantly by age category. The highest odds of infection by SARS-CoV-2 were seen in people between the ages of 35–44 (AOR = 1.33; 95% CI = 1.1, 1.67), 45–54 (AOR = 1.62; 95% CI = 1.28, 2.05), 55–64 (AOR = 1.78; 95% CI = 1.35, 2.35), and 65 and over years (AOR = 1.64; 95% CI = 1.21, 2.24) compared with those between the ages of 15–24 years.

This aligns with previous studies reporting higher seroprevalence among elder (50 years and above) as compared to younger age categories [6,38,39]. One study reported however that those under 50 years were the most affected by SARS-CoV-2 (26). Our findings are congruous with previous studies indicating that older adults may be more likely to exhibit symptoms of illness, or more likely to seek care [43]. Interestingly, some studies reported that seroprevalence was similar across age groups with similar exposure and susceptibility between these groups [4,37,44]. In contrast, older individuals (50 years and above) had a lower risk of

infection than younger individuals (12–29 years old), according to a study by Sizulu Moyo et al [45]. In Zambia, seroprevalence was lowest among those between the ages of 15 and 19, whereas in South Africa's Gauteng area, a seroprevalence survey revealed no age-related variations in seroprevalence [46,47]. A systematic review and meta-analysis of standardized population-based studies reported that children 0–9 years old and adults 60 years and older were at lower risk of seropositivity than adults 20 to 29 years old respectively, with relatively similar levels of infections in those aged 10–19 years and those 30–39, 40–49 and 50–59 years old, and in France seroprevalence was also higher in those aged 20–29 years old than in those 50-years and older [48,49].

In this study, SARS-CoV-2 seroprevalence was similar despite varying family size or smoking status, in contrast to other studies finding higher seroprevalence for instance in Leicester, UK, in larger HH size [50] or in smokers [42].

One strength of this study is that it is the first nation-wide population-based study to assess SARS-CoV-2 seroprevalence with a representative sample using a pre-in house validated antibody test method. This. These data have policy and practice implications. Low levels of seroconversion, far from those required for herd immunity, suggest that mass vaccinations is needed.

Limitations include single cross-sectional study design. The temporal causality cannot be determined between factors shown to be associated with seropositivity. The assay used to detect antibodies against SARS-CoV-2 is qualitative and it does not provide quantitative values, limiting the ability to estimate immunological status in the population. In addition, some of the given primary EA was not accessible due to security reason, which was latter substituted by CSA.

## Conclusion

In conclusion, our sero-study indicates nearly one in 10 individuals in Ethiopia was exposed to SARS-CoV-2 infection, which is more widespread than detected using PCR-based testing strategies so far. All regional states in the country have been affected by SARS-CoV-2 infection, although infection was more common in densely populated areas. Urban residents, those age 55-64 years, residing in Harari region, residing in Addis Ababa city administration, those with professional/technical occupations, and those having at least one comorbidity were found to be significantly more likely to have SARS-CoV-2 antibodies.

Therefore, epidemic prevention and control measures, efforts to identify and isolate new cases, testing and vaccine deployment to protect the vulnerable and reduce further spread should be strengthened until vaccine-driven herd immunity is reached. Individual infection prevention measures should be reinforced. Responsible authorities in each locality should prepare to ensure the health system is ready for a continued pandemic as the majority of the population is still at risk. It is also important to tailor public health interventions toward highly affected groups. Repeated surveys are planned for the future.

## Acknowledgement

This work was coordinated by Ethiopian Public Health Institute through the support of Federal Ministry of Health (FMoH) of Ethiopia, Ohio State University, US- Centers for Disease Control and prevention (US-CDC)-Ethiopia and Atlanta HQ offices, and Africa Centers for Disease Control and Prevention (ACDC) as well as World Health Organization (WHO). We are very much grateful to the National Research Task forces at FMoH for encouraging advice and guidance during this study period. We also acknowledge Central Statistics Agency (CSA), data collectors, supervisors and regional coordinators for their great contribution for quality data collection. All individuals who contributed for the success of this study, and study participants are highly acknowledged.

## Author contributions

**Conceptualization:** Birra Bejiga, Saro Abdella, Sarah Legare, Masresha Tessema, Ryan E. Tokarz, Sisay Alemayehu, S. Cornelia Kaydos-Daniels, David Sugerman, Berhanu Amare, Abraham Ali, Legesse Negash, Jemal Ayalew, Enyew Birru, Getachew Tollera, Mohammed Ahmed Rameto, Adamu Tayachew Mokennen, Dereje Nigussie, Laura E. Binkley, Joan-Miquel Balada-Llasat, Shu-Hua Wang, Hiwot Moges, Bernard Barekye, Marguerite Massinga Loembe, Frehywot Eshetu, Mohamed Abdulaziz, Wondwossen Gebreyes, Abebaw Gebeyehu, Getnet Yimer, Williams, Desmond E.

**Data curation:** Birra Bejiga, Sarah Legare, Amen Ben Hamida, Feyiso Bati, Legesse Negash, Jemal Ayalew, Frehywot Eshetu, Williams, Desmond E.

**Formal analysis:** Birra Bejiga, Amen Ben Hamida, Legesse Negash, Jemal Ayalew.

**Funding acquisition:** Birra Bejiga, Geremew Tasew Guma, Sisay Alemayehu, Frehywot Eshetu, Mohamed Abdulaziz, Faiqa K Ebrahim, Wondwossen Gebreyes, Williams, Desmond E.

**Investigation:** Birra Bejiga, Geremew Tasew Guma, Saro Abdella, Sarah Legare, Masresha Tessema, Dereje Nigussie, Joan-Miquel Balada-Llasat, Marguerite Massinga Loembe.

**Methodology:** Birra Bejiga, Feyiso Bati, Ryan E. Tokarz, Legesse Negash, Jemal Ayalew, Adamu Tayachew Mokennen, Eshetu Ejeta.

**Project administration:** Birra Bejiga, Geremew Tasew Guma, Masresha Tessema, Abraham Ali, Getachew Tollera, Leuel Lisanwork, Zelalem H. Mekuria, Frehywot Eshetu, Faiqa K Ebrahim, Wondwossen Gebreyes, Abebaw Gebeyehu, Dereje Duguma, Williams, Desmond E.

**Resources:** Birra Bejiga, Geremew Tasew Guma, Masresha Tessema, Feyiso Bati, Ryan E. Tokarz, Sisay Alemayehu, S. Cornelia Kaydos-Daniels, David Sugerman, Berhanu Amare, Abraham Ali, Mohammed Ahmed Rameto, Laura E. Binkley, Joan-Miquel Balada-Llasat, Shu-Hua Wang, Leuel Lisanwork, Zelalem H. Mekuria, Bernard Barekye, Frehywot Eshetu, Faiqa K Ebrahim, Getnet Yimer, Williams, Desmond E.

**Software:** Birra Bejiga, Amen Ben Hamida, Jemal Ayalew.

**Supervision:** Birra Bejiga, Geremew Tasew Guma, Saro Abdella, Feyiso Bati, Jemal Ayalew, Enyew Birru, Getachew Tollera, Leuel Lisanwork, Frehywot Eshetu, Eshetu Ejeta, Lia Tadsse, Getnet Yimer.

**Validation:** Birra Bejiga, Sarah Legare, Amen Ben Hamida, Legesse Negash, Shu-Hua Wang, Williams, Desmond E.

**Visualization:** Birra Bejiga, Eshetu Ejeta.

**Writing – original draft:** Birra Bejiga, Geremew Tasew Guma, Saro Abdella, Enyew Birru, Dereje Nigussie, Hiwot Moges, Frehywot Eshetu, Eshetu Ejeta, Williams, Desmond E.

**Writing – review & editing:** Birra Bejiga, Geremew Tasew Guma, Saro Abdella, Masresha Tessema, Eshetu Ejeta, Williams, Desmond E.

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
