## [Decision Letter · Decision Letter 0]

3 Jun 2024

PONE-D-24-10224Sero-prevalence of SARS-CoV-2 antibodies in Ethiopia: Results of the National Population Based Survey, 2021PLOS ONE

Dear Dr. Bedassa,

Thank you for submitting your manuscript to PLOS ONE. After careful consideration, we feel that it has merit but does not fully meet PLOS ONE’s publication criteria as it currently stands. Therefore, we invite you to submit a revised version of the manuscript that addresses the points raised during the review process.

We look forward to receiving your revised manuscript.

Kind regards,

Abraham Aregay Desta, MSc.

Academic Editor

PLOS ONE

2. Please provide additional details regarding participant consent. Since your study included minors, state whether you obtained consent from parents or guardians. If the need for consent was waived by the ethics committee, please include this information

4. We note that Figures 2, 3 , and 4 in your submission contain [map/satellite] images which may be copyrighted. All PLOS content is published under the Creative Commons Attribution License (CC BY 4.0), which means that the manuscript, images, and Supporting Information files will be freely available online, and any third party is permitted to access, download, copy, distribute, and use these materials in any way, even commercially, with proper attribution. For these reasons, we cannot publish previously copyrighted maps or satellite images created using proprietary data, such as Google software (Google Maps, Street View, and Earth). For more information, see our copyright guidelines: http://journals.plos.org/plosone/s/licenses-and-copyright.

a. You may seek permission from the original copyright holder of Figures 2, 3 , and 4 to publish the content specifically under the CC BY 4.0 license.  

Reviewers' comments:

Reviewer's Responses to Questions

**Comments to the Author**

1. Is the manuscript technically sound, and do the data support the conclusions?

Reviewer #1: Yes

Reviewer #2: Yes

Reviewer #3: Partly

2. Has the statistical analysis been performed appropriately and rigorously? 

Reviewer #1: No

Reviewer #2: Yes

Reviewer #3: No

3. Have the authors made all data underlying the findings in their manuscript fully available?

Reviewer #1: No

Reviewer #2: Yes

Reviewer #3: Yes

4. Is the manuscript presented in an intelligible fashion and written in standard English?

Reviewer #1: Yes

Reviewer #2: Yes

Reviewer #3: Yes

5. Review Comments to the Author

Reviewer #1: The study is important for decision making of the vaccine strategy and also understanding of the disease. i congratulate teh authors.

however, it is expected to provide following clarifications

1. sample size estimation is not clearly mentioned

2. what was the random selection method for individual selection within household ?

how much non response was considered ?

3. statistical methods adopted is not complete

4. no multivariate analysis is done , it is better to do that and discuss the findings. if not done for any particular reason, better to discuss and justify them.

Reviewer #2: This is an important study becuas if it is the first population-based inquiry in Ethippia.The manuscript is written in a way that makes for ease of understianding. However,I have the following obbservations/suggestions:

1. line66-69: Link the sentences together to the citation

2. line 70: Puncutation at the end of the sentence

3. line89-94: link snetences to citation

4. 268-270: I suggest you add prevalence figure to the age. This is will aid understanding.

5. Line 290: The statement "nearly one in ten" is difficult to comprrehend. You may like to use actual prevalnce here

6. I understad the style of putting the tables and figures as appendages, but I suggest putting them in the body og the writing, unless the will be highlighted as links.

Reviewer #3: Abstract

The writers have properly captured the essence of the work by highlighting the key research question, the method(s) used and their key findings in a style that is clear, easy to read and comprehend.

 Concerns:

1. Conclusion and recommendation: You already have your result from the study, rather than draw your conclusion on "the limited knowledge on SARSCoV-2 seroprevalence", connect your conclusion to the prevalence levels observed in your study and the implications of same.

2. What is the antibodies majority of your population as you indicated on line 52 of your submission?

Introduction

The writers provide a very good background on the subject including highlighting some interesting hypothesis on why the pandemics spared Africa as well as the a good argument for using serosurveys to determine the true extent of the pandemic considering the use-case limitation to the RT-PCR...how it misses asymptomatic and mildly symptomatic people who may choose not to attend the hospital for care or may not be considered for testing with RT-PCR at the triage.

concerns:

1. In lines 68 and 69, you need to specifically say that the number of REPORTED SARSCoV-2 infections could be underestimating the true number of infections sin your study population

2. Also, I feel that the conclusive part of your introduction could help recast the conclusion in your abstract to relate more to your findings and its implications

Methodology

concerns

1. what is the justification for using 16years as the lower boundary for age in your study?

2.On line 133, confirm which EA size parameter was used in the first stage. Was it HH-size, HH-number, land mass? You need to be specific.

3. On line 154-6: Since the category of staff that conducted the RDT screening is not explicitly stated, is it safe to assume that the Data collectors and field supervisors who were trained are the ones who did the testing? Please what is their professional qualification?

4. Were the test performance characteristics of the RDT kit verified before deployed for this study?

5. How did you obtain written consent from all participants, were they all able to read and sign the document?

Result and Discussion

1. State the actual Pvalues where statistical tests were done and for all instance where differences were observed such as region, age etc please state the actual pvalue rather than "Pvalue 0.05".

2. Discuss implication of the SARSCoV2 IgM observation in your study

6. PLOS authors have the option to publish the peer review history of their article (what does this mean? ). If published, this will include your full peer review and any attached files.

**Do you want your identity to be public for this peer review?** For information about this choice, including consent withdrawal, please see our Privacy Policy .

Reviewer #1: **Yes: ** Sumanth Mallikarjuna Majgi

Reviewer #2: **Yes: ** Prince Obinna Anyanwu

Reviewer #3: No

---

## [Author Response · Author response to Decision Letter 1]

2 Sep 2024

5. Review Comments to the Author

Reviewer #1: The study is important for decision making of the vaccine strategy and also understanding of the disease. i congratulate teh authors.

Authors’ response: Thanks. Noted.

However, it is expected to provide following clarifications

I. Sample size estimation is not clearly mentioned

Authors’ response: These are now addressed in detail under subtopics, Sample size and sampling procedure.

II. What was the random selection method for individual selection within household?

Authors’ response: Random selection is a form of sampling where a representative group of research participants is selected from a larger group by chance. this is now described under (line 297-328).

III. How much non-response was considered?

Authors’ response: Paragraph (lines 176-179) revised non-response rate is included.

IV. Statistical methods adopted is not complete

Authors’ response: Paragraph (lines 223-247) revised as suggested under the Statistical analysis part.

V. No multivariate analysis is done, it is better to do that and discuss the findings. If not done for any particular reason, better to discuss and justify them.

Authors’ response: These are now addressed in detail under topics: Factors associated with COVID-19 Seroprevalence Results of a multilevel mixed effect logistic regression model

(lines 191-228).

Reviewer #2: This is an important study because if it is the first population-based inquiry in Ethiopia. The manuscript is written in a way that makes for ease of understanding. However, I have the following observations/suggestions:

Authors’ response: Thanks. Noted.

A. line66-69: Link the sentences together to the citation

Authors’ response: Done

B. line 70: Punctuation at the end of the sentence

Authors’ response: Done

C. line89-94: link sentences to citation

Authors’ response: Done

D. 268-270: I suggest you add prevalence figure to the age. This is will aid understanding.

Authors’ response: this is now added and corrected

E. Line 290: The statement "nearly one in ten" is difficult to comprehend. You may like to use actual prevalence here.

Authors’ response: This is now corrected and well in order.

F. I understand the style of putting the tables and figures as appendages, but I suggest putting them in the body og the writing, unless the will be highlighted as links.

Authors’ response: Formatted as suggested and cross-referenced by highlighting the link

Reviewer #3: Abstract

The writers have properly captured the essence of the work by highlighting the key research question, the method(s) used and their key findings in a style that is clear, easy to read and comprehend.

Authors’ response: Thanks. Noted.

Concerns:

1. Conclusion and recommendation: You already have your result from the study, rather than draw your conclusion on "the limited knowledge on SARSCoV-2 seroprevalence", connect your conclusion to the prevalence levels observed in your study and the implications of same.

Authors’ response: The conclusion is revised. The Point is well taken.

2. What is the antibodies majority of your population as you indicated on line 52 of your submission?

Authors’ response: Point well taken. We revised the Conclusion and recommendation section addressing the comments given here and by others.

Introduction

The writers provide a very good background on the subject including highlighting some interesting hypothesis on why the pandemics spared Africa as well as the a good argument for using serosurveys to determine the true extent of the pandemic considering the use-case limitation to the RT-PCR...how it misses asymptomatic and mildly symptomatic people who may choose not to attend the hospital for care or may not be considered for testing with RT-PCR at the triage.

Authors’ response: Thanks. Noted.

Concerns:

1. In lines 68 and 69, you need to specifically say that the number of REPORTED SARSCoV-2 infections could be underestimating the true number of infections sin your study population

Authors’ response: This is reviewed. We have fixed it now that we realized it needed to be connected to previous sentences and cited. Now corrected.

2. Also, I feel that the conclusive part of your introduction could help recast the conclusion in your abstract to relate more to your findings and its implications

Authors’ response: The conclusion section revised taking this comment and other to shorten and make the section crisper.

Methodology

concerns

1. what is the justification for using 16years as the lower boundary for age in your study?

Authors’ response: Published serological studies suggest that younger adults (particularly those aged above 15y) often have high cumulative rates of SARS-CoV-2 infection in the community. The covid-19 infection is much less in age less than 15 due to different scenarios. We undertook a literature review using the Living Evidence on COVID-19, a database collecting COVID-19 related published articles from Pubmed and EMBASE and preprints from medRxiv and bioRxiv, with MESH terms.

2. On line 133, confirm which EA size parameter was used in the first stage. Was it HH-size, HH-number, land mass? You need to be specific.

Authors’ response: Paragraph (lines 133) revised as suggested.

3. On line 154-6: Since the category of staff that conducted the RDT screening is not explicitly stated, is it safe to assume that the Data collectors and field supervisors who were trained are the ones who did the testing? Please what is their professional qualification?

Authors’ response: These are now addressed in detail under subtopics data collection and participant consent.

4. Were the test performance characteristics of the RDT kit verified before deployed for this study?

Authors’ response: Quality control (QC) panels consisting of a positive and negative control specimen were done in parallel with the testing procedure to ensure test kits were performing correctly. This is now indicated in the manuscript (lines 210-219).

5. How did you obtain written consent from all participants, were they all able to read and sign the document?

Authors’ response: Unfortunately, the majority of our study participants are literate; nevertheless, for those who are not, we followed the instructions for Illiterate Subjects while presenting the informed consent form orally.

Result and Discussion

1. State the actual P values where statistical tests were done and for all instance where differences were observed such as region, age etc please state the actual p value rather than "P value 0.05".

Authors’ response: Point well taken. Revised the result and discussion section addressing the comments given here and by others.

2. Discuss implication of the SARSCoV2 IgM observation in your study

Authors’ response: Done

---

## [Decision Letter · Decision Letter 1]

24 Nov 2024

PONE-D-24-10224R1Sero-prevalence of SARS-CoV-2 antibodies in Ethiopia: Results of the National Population Based Survey, 2021PLOS ONE

Dear Dr. Bedassa,

Thank you for submitting your manuscript to PLOS ONE. After careful consideration, we feel that it has merit but does not fully meet PLOS ONE’s publication criteria as it currently stands. Therefore, we invite you to submit a revised version of the manuscript that addresses the points raised during the review process.

We look forward to receiving your revised manuscript.

Kind regards,

Sk Md Mamunur Rahman Malik

Academic Editor

PLOS ONE

Reviewers' comments:

Reviewer's Responses to Questions

**Comments to the Author**

1. If the authors have adequately addressed your comments raised in a previous round of review and you feel that this manuscript is now acceptable for publication, you may indicate that here to bypass the “Comments to the Author” section, enter your conflict of interest statement in the “Confidential to Editor” section, and submit your "Accept" recommendation.

Reviewer #1: All comments have been addressed

Reviewer #2: (No Response)

Reviewer #4: (No Response)

2. Is the manuscript technically sound, and do the data support the conclusions?

Reviewer #1: Yes

Reviewer #2: (No Response)

Reviewer #4: Partly

3. Has the statistical analysis been performed appropriately and rigorously? 

Reviewer #1: Yes

Reviewer #2: (No Response)

Reviewer #4: No

4. Have the authors made all data underlying the findings in their manuscript fully available?

Reviewer #1: Yes

Reviewer #2: (No Response)

Reviewer #4: No

5. Is the manuscript presented in an intelligible fashion and written in standard English?

Reviewer #1: Yes

Reviewer #2: (No Response)

Reviewer #4: Yes

6. Review Comments to the Author

Reviewer #1: All the concerns have been addressed in all sections of manuscript , including sample size , sampling and recommendation

Reviewer #2: (No Response)

Reviewer #4: The statistical methodology of this study involved descriptive analysis, sampling using a complex survey design, and mixed-effects logistic regression analysis.

The descriptive analysis requires a table presenting the demographic and clinical characteristics of the study sample. It is important to note the distinction between the study sample and the broader sampled population. The current Table 1 describes the overall population, but no table provides a descriptive analysis of the study sample. This information is important to understand the sizes of subgroups analyzed in the logistic regression.

Another concern with Table 1 is that it combines seroprevalence levels estimated in the study with data on the national population obtained from the census. I strongly recommend separating these into two distinct tables. This will give a clear notion on which results are from the current study. Additionally, since the table on seroprevalence contains the same information as Figure 1, I say that Figure 1 should be removed. While Figure 1 is visually appealing, a table would be more useful for readers who need exact numerical estimates.

The estimation of the sample size is included in this version of the manuscript, but there are issues with the formula provided. The formula does not include a term representing the required level of confidence (usually expressed as a function of the z statistic). Furthermore, the factor accounting for prevalence is conventionally expressed as p(1-p) , but the formula in the manuscript uses (1/p - 1) . Therefore, this should be verified and have an explanation in a response. It is also worth noting that the current formatting makes it appear as though the "-1" is an exponent of p , which could be misleading.

Additionally, applying the formula with the numbers provided yields a sample size estimate of N = 21,495 , which differs from the value reported in the manuscript. Again more clarification is required.

Finally, the logistic regression appears to be well-conducted, and the results are presented effectively. However, it remains critical to include a table showing the frequencies (descriptive analysis) of the variables analyzed in the obtained sample. This would allow for the identification of potential biases in certain groups, which should be acknowledged in the Discussion section, if biases arise.

Additional comments below.

Abstract

Several sections of the text use "COVID-19" and "SARS-CoV-2" almost interchangeably. It is important to make distinction between the disease and the virus. Therefore, phrases such as “seroprevalence of COVID-19,” “infected by COVID-19,” and “spread of COVID-19” need revision to ensure the correct terminology is used throughout the manuscript.

Additionally, the name of the disease, COVID-19, should always be capitalized and used consistently across the text.

In the Abstract, it is stated that AOR (adjusted odds ratios) were used to identify independent associations. However, it might be more accurate to describe the findings as identifying groups at higher risk of having been infected at least once.

The reported general seroprevalence levels differ slightly between the Abstract and the Results section. I recommend verifying these values and ensuring consistency throughout the manuscript to avoid any discrepancies.

Introduction

- Sentences appear truncated in lines 85–88.

- Line 90 (sensitivity of the test): Specify which test is being referred to in this context.

Methods

- Define "HH" upon first use.

- Line 167 : Replace d with \delta.

- Clarify how the design effect (DEFT) was estimated at 2.25.

- As mentioned, provide more explanation on the sample size formula and verify it.

Results

- As mentioned earlier, a table presenting the demographic and clinical information of the study sample should be included.

- As previously noted, Figure 1 should be removed since the seroprevalence data it displays is already provided in a table.

- If the proportion of individuals with no exposure to either IgG or IgM is reported as 90.7%, then the exposure rate should be 9.3%. However, the abstract states this as 10%. This discrepancy needs to be addressed to ensure clarity and consistency throughout the manuscript, including in the abstract.

- Table 3: phrase "ever sicked" requires correction.

Discussion

- I think that finding non-significant results about household size requires discussion. This is yet another reason why the table with characteristics of the sample is important.

7. PLOS authors have the option to publish the peer review history of their article (what does this mean? ). If published, this will include your full peer review and any attached files.

**Do you want your identity to be public for this peer review?** For information about this choice, including consent withdrawal, please see our Privacy Policy .

Reviewer #1: **Yes: ** Sumanth Mallikarjuna Majgi

Reviewer #2: **Yes: ** Prince Obinna Anyanwu

Reviewer #4: No

---

## [Author Response · Author response to Decision Letter 2]

30 Dec 2024

Nov 24 2024

PONE-D-24-10224 PONE-D-24-10224R1

Manuscript Title: Sero-prevalence of SARS-CoV-2 antibodies in Ethiopia: Results of the National Population Based Survey, 2021

Dear Editor,

On behalf of the authors, and myself I would like to thank you for and the reviewers for having critically reviewed the manuscript and for giving allowing us to revise and re-resubmit it to PLOS ONE. We have revised the manuscript based on the comments and suggestions from you and all three reviewers.

We deeply appreciate yours and the reviewers' comments and suggestions, which we found very helpful and used to improve the quality of the manuscript. Per your advice, we have prepared a point-by-point response to all comments as presented below. We have also made changes as advised in a separate file titled 'Revised Manuscript with Track Changes' and have uploaded the file per PLOS guidelines. In addition, we have uploaded three documents as part of our online submission: the tracked and clean copies of the manuscript as well as a rebuttal letter.

Sincerely

Birra Bejiga

Reviewer's Responses to Questions

Comments to the Author

1. If the authors have adequately addressed your comments raised in a previous round of review and you feel that this manuscript is now acceptable for publication, you may indicate that here to bypass the “Comments to the Author” section, enter your conflict of interest statement in the “Confidential to Editor” section, and submit your "Accept" recommendation.

Reviewer #1: All comments have been addressed

Reviewer #2: (No Response)

Reviewer #4: (No Response)

2. Is the manuscript technically sound, and do the data support the conclusions?

Reviewer #1: Yes

Reviewer #2: (No Response)

Reviewer #4: Partly

3. Has the statistical analysis been performed appropriately and rigorously?

Reviewer #1: Yes

Reviewer #2: (No Response)

Reviewer #4: No

4. Have the authors made all data underlying the findings in their manuscript fully available?

Reviewer #1: Yes

Reviewer #2: (No Response)

Reviewer #4: No

5. Is the manuscript presented in an intelligible fashion and written in standard English?

Reviewer #1: Yes

Reviewer #2: (No Response)

Reviewer #4: Yes

6. Review Comments to the Author

Reviewer #1: All the concerns have been addressed in all sections of manuscript, including sample size, sampling and recommendation.

Authors’ response: Thanks. Noted.

Reviewer #2: (No Response):

Reviewer #4: Abstract

The statistical methodology of this study involved descriptive analysis, sampling using a complex survey design, and mixed-effects logistic regression analysis.

Authors’ response: Thanks. Noted.

The descriptive analysis requires a table presenting the demographic and clinical characteristics of the study sample. It is important to note the distinction between the study sample and the broader sampled population. The current Table 1 describes the overall population, but no table provides a descriptive analysis of the study sample. This information is important to understand the sizes of subgroups analyzed in the logistic regression.

Authors’ response:

Many thanks for your reflection. Lines 611 Table 1, Column 3, labelled ‘Pop. Distribution' indicates the weighted percentage distribution of the demographic and clinical variables. This approach is the recommended data analysis presentation, as it adjusts for survey weights, rather than simply displaying descriptive statistics without such adjustment. However, we revised it with some correction.

Another concern with Table 1 is that it combines seroprevalence levels estimated in the study with data on the national population obtained from the census. I strongly recommend separating these into two distinct tables. This will give a clear notion on which results are from the current study. Additionally, since the table on seroprevalence contains the same information as Figure 1, I say that Figure 1 should be removed. While Figure 1 is visually appealing, a table would be more useful for readers who need exact numerical estimates.

Authors’ response:

Thank you for your critical review. As per the reviewer's suggestion, lines 593 revised as suggested we have removed Figure 1, as it depicts similar information to that presented in Table 1. However we did not use census data in this analysis; instead, we applied survey weights to the sample of 12,756, which is crucial for adjusting of over and under sampling estimation. This sample represents a population of 19,456,360 in Ethiopia through the application of survey weights. Therefore, we combined both pieces of information into a single table, as suggested.

The estimation of the sample size is included in this version of the manuscript, but there are issues with the formula provided. The formula does not include a term representing the required level of confidence (usually expressed as a function of the z statistic). Furthermore, the factor accounting for prevalence is conventionally expressed as p(1-p) , but the formula in the manuscript uses (1/p - 1) . Therefore, this should be verified and have an explanation in a response. It is also worth noting that the current formatting makes it appear as though the "-1" is an exponent of p , which could be misleading.

Authors’ response:

Line 163, revised and added as suggested. However, the sample size calculation formula we used was the national sample size determination formula proposed and utilized by the Demographic and Health Survey group. This formula is derived from the p (1−p) formula, which is suitable for national surveys and particularly favours for small p values. The p (1−p) formula is mostly applicable for small-scale surveys and single population proportions.

Additionally, applying the formula with the numbers provided yields a sample size estimate of N = 21,495, which differs from the value reported in the manuscript. Again, more clarification is required.

Authors’ response: Point well taken. We made the necessary corrections and adjustments to the sample size and sample size formula the details are now given on (lines 141-172) the confusion arose unintentionally. We apologize for any confusion this has created.

Finally, the logistic regression appears to be well conducted, and the results are presented effectively. However, it remains critical to include a table showing the frequencies (descriptive analysis) of the variables analyzed in the obtained sample. This would allow for the identification of potential biases in certain groups, which should be acknowledged in the Discussion section, if biases arise.

Authors’ response:

Thank you Point well taken. However, it was indicated on table 1 column 3 on (line 612) the weighted percentage distribution of the demographic and clinical variables. We combine both the descriptive statistics for each variable and the corresponding SARS-CoV-2 prevalence into one Table to reduce the number of Tables.

Additional comments below.

Abstract

Several sections of the text use "COVID-19" and "SARS-CoV-2" almost interchangeably. It is important to make distinction between the disease and the virus. Therefore, phrases such as “seroprevalence of COVID-19,” “infected by COVID-19,” and “spread of COVID-19” need revision to ensure the correct terminology is used throughout the manuscript.

Authors’ response: thank you revised as suggested.

Additionally, the name of the disease, COVID-19, should always be capitalized and used consistently across the text.

Authors’ response: thank you corrected and edited.

In the Abstract, it is stated that AOR (adjusted odds ratios) were used to identify independent associations. However, it might be more accurate to describe the findings as identifying groups at higher risk of having been infected at least once.

Authors’ response: Thank you, revised and added as suggested.

The reported general seroprevalence levels differ slightly between the Abstract and the Results section. I recommend verifying these values and ensuring consistency throughout the manuscript to avoid any discrepancies.

Authors’ response: This is now corrected and well in order

Introduction

- Sentences appear truncated in lines 85–88.

Authors’ response: This is now complete

- Line 90 (sensitivity of the test): Specify which test is being referred to in this context.

Authors’ response: thank you Revised as suggested.

Methods

- Define "HH" upon first use.

Authors’ response: thank you Revised as suggested.

- Line 167 : Replace d with \delta.

Authors’ response: Thank you replaced and edited as suggested.

- Clarify how the design effect (DEFT) was estimated at 2.25.

Authors’ response: Many thanks we have identified that it was not correctly wrote. This in now corrected and it all makes sense.

- As mentioned, provide more explanation on the sample size formula and verify it.

Authors’ response:

The sample size formula was taken from the Demographic and Health survey manual which is developed from the usual sample size formula p(1-p) that is planned for national sample size determination to favor the small proportions.

Results

- As mentioned earlier, a table presenting the demographic and clinical information of the study sample should be included.

Authors’ response: Thank you Point well taken. However, it was indicated on table 1 column 3 on (line 612) the weighted percentage distribution of the demographic and clinical variables. We combine both the descriptive statistics for each variable and the corresponding SARS-CoV-2 prevalence into one Table to reduce the number of Tables.

- As previously noted, Figure 1 should be removed since the seroprevalence data it displays is already provided in a table.

Authors’ response: Deleted as suggested

- If the proportion of individuals with no exposure to either IgG or IgM is reported as 90.7%, then the exposure rate should be 9.3%. However, the abstract states this as 10%. This discrepancy needs to be addressed to ensure clarity and consistency throughout the manuscript, including in the abstract.

Authors’ response: This is reviewed we have identified that it was not consistence. This in now corrected and it all makes sense.

- Table 3: phrase "ever sicked" requires correction.

Authors’ response: Done

Discussion

- I think that finding non-significant results about household size requires discussion. This is yet another reason why the table with characteristics of the sample is important.

Authors’ response: Noted

7. PLOS authors have the option to publish the peer review history of their article (what does this mean?). If published, this will include your full peer review and any attached files.

Do you want your identity to be public for this peer review? For information about this choice, including consent withdrawal, please see our Privacy Policy.

Reviewer #1: Yes: Sumanth Mallikarjuna Majgi

Reviewer #2: Yes: Prince Obinna Anyanwu

Reviewer #4: No

---

## [Decision Letter · Decision Letter 2]

21 Jan 2025

Sero-prevalence of SARS-CoV-2 antibodies in Ethiopia: Results of the National Population Based Survey, 2021

PONE-D-24-10224R2

Dear Dr. Bedassa,

We’re pleased to inform you that your manuscript has been judged scientifically suitable for publication and will be formally accepted for publication once it meets all outstanding technical requirements.

Kind regards,

Sk Md Mamunur Rahman Malik

Academic Editor

PLOS ONE

Additional Editor Comments (optional):

Reviewers' comments:

Reviewer's Responses to Questions

**Comments to the Author**

1. If the authors have adequately addressed your comments raised in a previous round of review and you feel that this manuscript is now acceptable for publication, you may indicate that here to bypass the “Comments to the Author” section, enter your conflict of interest statement in the “Confidential to Editor” section, and submit your "Accept" recommendation.

Reviewer #4: (No Response)

2. Is the manuscript technically sound, and do the data support the conclusions?

Reviewer #4: Yes

3. Has the statistical analysis been performed appropriately and rigorously? 

Reviewer #4: Yes

4. Have the authors made all data underlying the findings in their manuscript fully available?

Reviewer #4: Yes

5. Is the manuscript presented in an intelligible fashion and written in standard English?

Reviewer #4: Yes

6. Review Comments to the Author

Reviewer #4: Regarding the sample size formula, I realize that the relative standard error was used, which explains the formula. I should note that in my understanding the value of 4.8% for delta was arbitrary. Furthermore, the text still refers to the margin of error (line 160) instead of the relative measure. I strongly suggest indicating the relative standard error in the text for consistency.

Table 3: For the Afar region as the reference (Model 3), the value should be 1.

Table 1: My recommendation was to separate the results on the estimated seroprevalence levels into a separate table. I strongly favor this presentation, however the final decision lies with the authors.

Table 1: It should be made clear that the population column was evaluated using a weighted approach.

Line 40: Symptoms compatible with SARS-CoV-2 infection

7. PLOS authors have the option to publish the peer review history of their article (what does this mean? ). If published, this will include your full peer review and any attached files.

**Do you want your identity to be public for this peer review?** For information about this choice, including consent withdrawal, please see our Privacy Policy .

Reviewer #4: No

---

## [Editor Report · Acceptance letter]

PONE-D-24-10224R2

PLOS ONE

Dear Dr. Bejiga,

I'm pleased to inform you that your manuscript has been deemed suitable for publication in PLOS ONE. Congratulations! Your manuscript is now being handed over to our production team.

Kind regards,

on behalf of

Dr. Sk Md Mamunur Rahman Malik

Academic Editor

PLOS ONE